# Three-component radical homo Mannich reaction

Shuai Shi[1], Wenting Qiu[1], Pannan Miao[1], Ruining Li[1], Xianfeng Lin[1] & Zhankui Sun [1]✉

Aliphatic amine, especially tertiary aliphatic amine, is one of the most popular functionalities found in pharmaceutical agents. The Mannich reaction is a classical and widely used transformation for the synthesis of β-amino-carbonyl products. Due to an ionic nature of the mechanism, the Mannich reaction can only use non-enolizable aldehydes as substrates, which significantly limits the further applications of this powerful approach. Here we show, by employing a radical process, we are able to utilize enolizable aldehydes as substrates and develop the three-component radical homo Mannich reaction for the streamlined synthesis of γ-amino-carbonyl compounds. The electrophilic radicals are generated from thiols via the desulfurization process facilitated by visible-light, and then add to the electron-rich double bonds of the in-situ formed enamines to provide the products in a single step. The broad scope, mild conditions, high functional group tolerance, and modularity of this metal-free approach for the synthesis of complex tertiary amine scaffolds will likely be of great utility to chemists in both academia and industry.

[1] Shanghai Key Laboratory for Molecular Engineering of Chiral Drugs, School of Pharmacy, Shanghai Jiao Tong University, No. 800 Dongchuan Rd., 200240 Shanghai, China. ✉email: zksun@sjtu.edu.cn

Amines are very important functional groups in medicinal chemistry and are present in many drugs[1–3]. They may be involved in H-bonding with target binding sites, either as hydrogen-bond acceptors or hydrogen-bond donors. In many cases, an amine could be protonated and a strong ionic interaction may take place with electron-negative part in the binding site[4]. Therefore, there is no doubt that aliphatic amine, especially tertiary aliphatic amine, is one of the most popular functional groups found in pharmaceutical agents[2]. Despite their importance, current synthetic techniques for amines are still limited[5–17]. The development of mild, modular and efficient synthesis of amines is still in pressing need.

The Mannich reaction is a classical reaction for the synthesis of β-amino-carbonyl products[18–23]. It has been known for more than a century and is widely used in many areas of organic chemistry[24–32]. It has also been frequently proposed in many biosynthetic pathways, especially for alkaloids biosynthesis[33,34]. This reaction utilizes a non-enolizable aldehyde, a secondary amine and an enolizable carbonyl compound as starting materials, and affords useful β-amino-carbonyl products in one step. The use of non-enolizable aldehyde is essential to form the Schiff base intermediate, which acts as an electrophile and reacts with the enolizable carbonyl compound to provide the Mannich product. However, for the enolizable aldehyde, an electron-rich enamine intermediate will form, which will not

react with the enolizable carbonyl compound (Fig. 1a). Therefore, the Mannich reaction is mainly for non-enolizable aldehydes. This obvious limitation poses great challenge for the further applications of this elegant and powerful transformation.

Seeking to overcome this obstacle, we hypothesized that the addition of an electrophilic radical **I** to the double bond of the electron-rich enamine would match the polarity request for radical reactions and generate radical intermediate **II** (Fig. 1b)[35]. This radical intermediate **II** could be stabilized by the adjacent nitrogen and then be intercepted through hydrogen atom transfer to provide γ-amino-carbonyl compound in a single step, while this type of products could not easily be accessed by other methods. To the best of our knowledge, this radical homo Mannich reaction has not been realized yet. There are several challenges. First of all, this radical-based reaction requires the use of mild conditions to selectively generate the electrophilic radical while not affecting other sensitive substances, such as the aldehyde, the enamine intermediate and the γ-amino-carbonyl product. These compounds could be reactive under radical conditions. Furthermore, the hydrogen atom transfer must be capable of rapidly intercepting radical intermediate **II** while not quenching the electrophilic radical **I**. Thirdly, other side reactions, such as the aldol reaction should be avoided under this condition. In spite of these challenges, herein we report the realization of our

**Fig. 1 Radical strategies for homo Mannich reaction. a** The classical Mannich reaction. **b** Radical homo Mannich reaction. **c** Proposed mechanism for the three-component radical homo Mannich reaction. HAT hydrogen atom transfer.

**Fig. 2 Scope of the radical homo Mannich reaction. a** The model reaction. **b** Optimal reaction conditions. DTBP Di-tert-butyl peroxide, MS molecular sieves. **c** Scope of the amine partner. [a]triethyl phosphite (1.5 equiv.) was used instead of PPh₃. [b]the dr value was based on ¹H NMR.

hypothesis through the development of three-component radical homo Mannich reaction for the streamlined synthesis of complex tertiary amines. We chose to generate the electrophilic radical **I** through desulfurization of thiols[36,37]. We believe the mild conditions could tolerate different functionalities. Besides, the thiol itself is an excellent hydrogen atom transfer reagent to intercept radical intermediate **II**[38]. The mild reaction conditions could also suppress other side reactions. Thus, based on our proposal, a possible mechanism is depicted in Fig. 1c. During the preparation of this paper, a multicomponent strategy for the construction of β-trifluoromethylated tertiary alkylamines was reported[35].

Here we show, by employing a radical process, we are able to expand the scope of classical Mannich reaction to enolizable aldehydes for the streamlined synthesis of γ-amino-carbonyl compounds.

## Results

**Optimization studies.** We started our model reaction (Fig. 2a) using 3-phenylpropanal (**1**), N-methyl-1-phenylmethanamine (**2**), and ethyl 2-mercaptoacetate (**3**). Gratifyingly, this reaction worked perfectly well in DCM with 4 Å molecular sieves and provided the desired product (**4**) in 91% isolated yield within

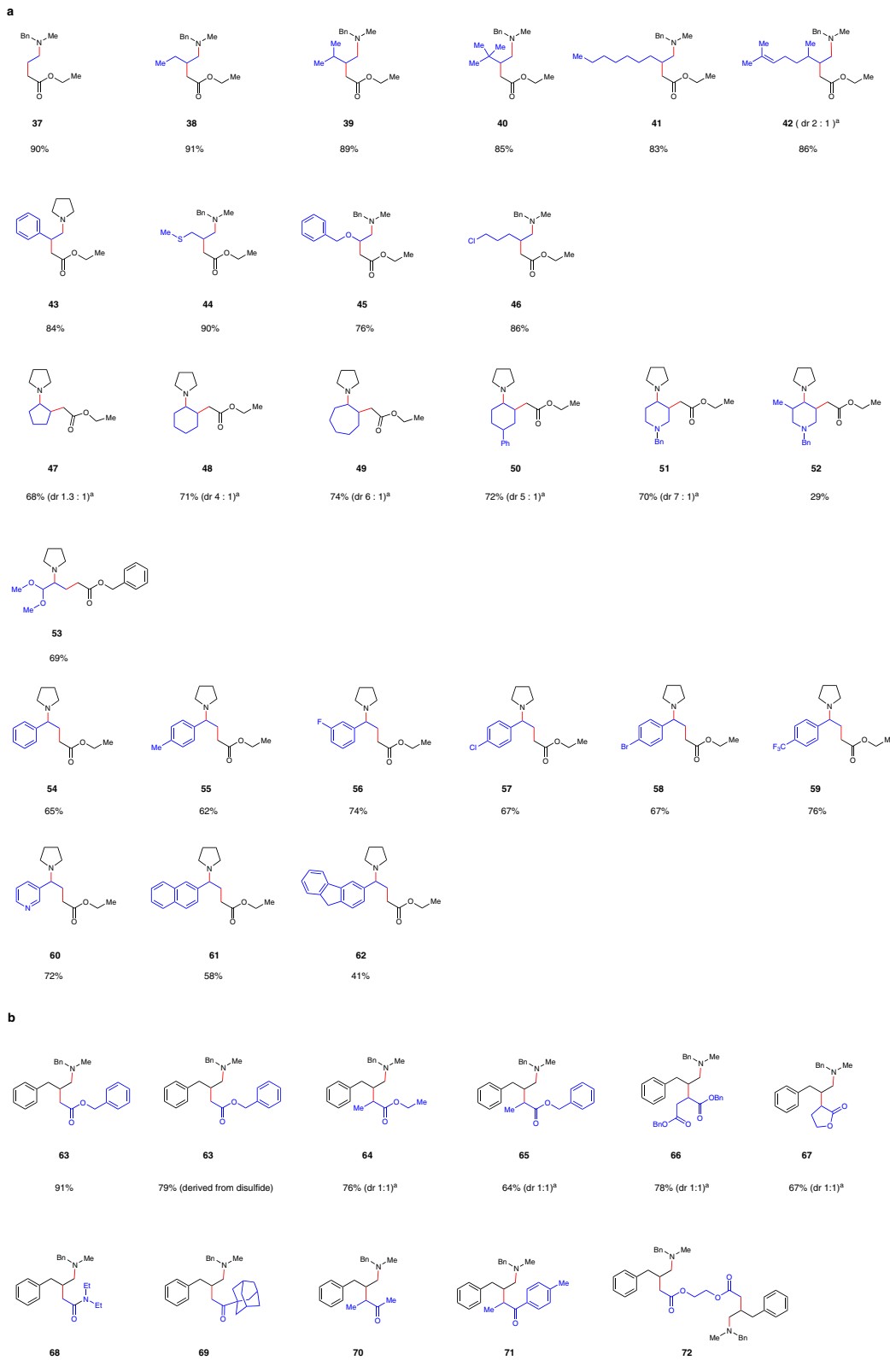

**Fig. 3 Scope of the radical homo Mannich reaction. a** Scope of the aldehyde and ketone partner. **b** Scope of the thiol partner. [a]the dr value was based on ¹H NMR.

10 h under visible-light. Further experiments demonstrated that it could tolerate different solvents and different phosphoric reagents (for a detailed account of the optimization study, see Supplementary Table 1). The reaction could also be performed in greener solvent[39], such as ethyl acetate. Control experiments revealed no reactions occurred in the absence of phosphoric reagent, DTBP (di-*tert*-butyl peroxide), or visible-light.

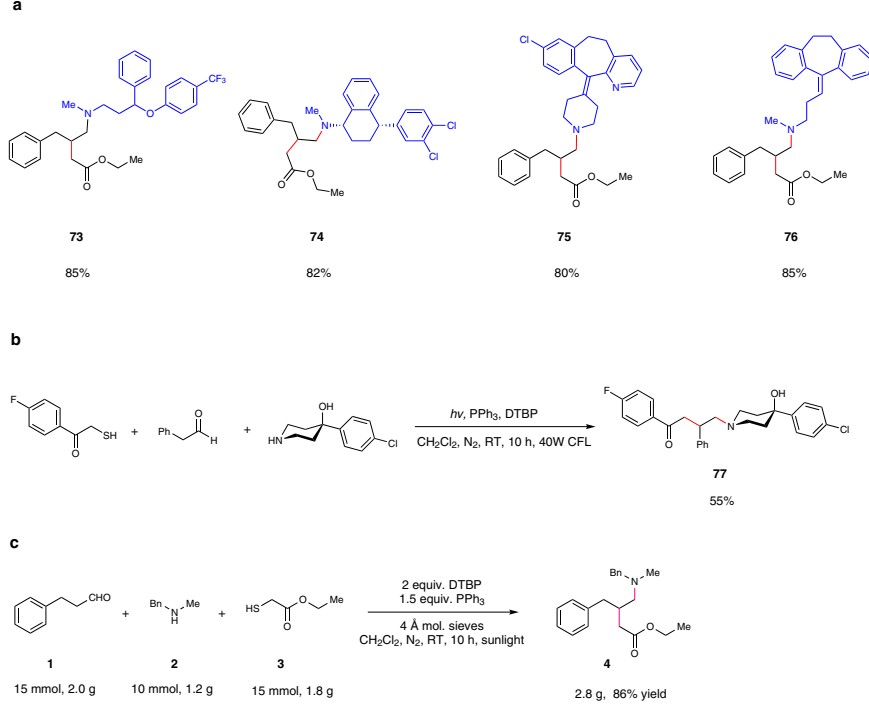

**Fig. 4 Synthetic utilities of the radical homo Mannich reaction. a** Late-stage modification of pharmaceutical agents. **b** Synthesis of the analog of Haloperidol. **c** Scale-up reaction for compound **4** under the sunlight.

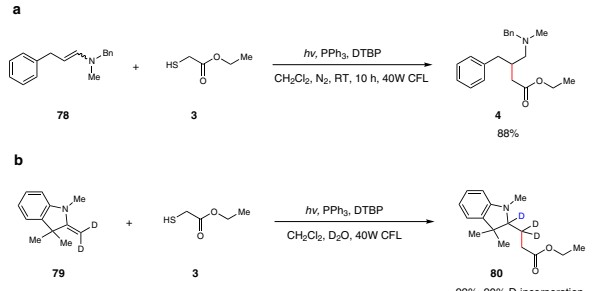

**Fig. 5 Experimental observations for the proposed mechanism. a** Direct addition to the enamine. **b** Deuterium-labeling study.

**Evaluation of substrate scope.** Having established the optimized conditions (Fig. 2b), we started to probe the scope of this transformation. We first evaluated this method with different amines (Fig. 2c). Both linear and cyclic amines worked well. A broad range of functional groups such as benzyl group (**4, 14-16**), *para*-methoxy benzyl group (**5 and 17**), aryl fluoride (**6**), aryl bromide (**7**), pyridine (**13**), azetidine (**18**), pyrrolidine (**19-21**), piperidine (**22-28, 33**), tertiary amine (**25**), ether (**28**), piperazine (**29**), benzyloxycarbonyl (**30**), thiomorpholine (**31**), and morpholine (**32**), could be well tolerated and provided the products in good to excellent yields. More sensitive functionalities which could be reactive under normal Mannich reaction conditions, such as ester (**14, 15 and 24**), cyanide (**10 and 27**), bromide (**26**), and even free hydroxy group (**33**), proved to be compatible and furnished the products in good to excellent yields, reflecting the mildness of the reaction conditions. As for substrate **9** with a terminal alkene, the radical added to the enamine rather than the alkene, probably due to the electron-rich nature of the enamine intermediate. Less reactive anilines could also be successfully transformed into the corresponding products in good yields (**34 and 35**) when triethyl

phosphite was used instead of PPh₃. However, with sterically more hindered aniline substrate (**36**), the yield was unsatisfactory.

The scope of the aldehyde partner was then examined (Fig. 3a). Various aldehydes proceeded efficiently to give the products in good to excellent yields. Notable examples included the ones bearing alkene (**42**), sulfide (**44**), ether (**45**), benzyl (**45**), and even chloride group (**46**).

Ketones also proved to be suitable substrates. Different sized cyclic ketones worked efficiently (**47-49**). Benzyl protected piperidin-4-one also reacted well (**51**). For unsymmetric ketones, the reaction took place at the less hindered sites (**52-53**).

Aryl methyl ketone with different substituents, either electron-donating groups (**55**) or electron-withdrawing groups (**56-59**), proceeded to afford the desired products. Pyridine substrate also provided the product in good yield (**60**).

We next surveyed the scope of thiols (Fig. 3b). Different esters (**63-66**), lactones (**67**) and amides (**68**) bearing α-sulfide were efficiently converted to the corresponding products in good to excellent yields. It is worthy pointing out that disulfides could be compatible partners as well (**63**). α-Substituted sulfide ketones were also readily accommodated (**70-71**). When there are two reacting sites, we were able to isolate the bi-functionalized product in 72% yield (**72**).

**Synthetic utilities.** In a final effort to establish the generality of this protocol, we did a late-stage modification of highly functionalized commercial drugs. As dialkylamine motifs are present in a range of small-molecule drugs and pre-clinical candidates, we selected four pharmaceutical agents and subjected them to the radical homo Mannich reactions (Fig. 4a). Each of these structurally complex amines underwent smooth transformations and furnished the tertiary amine products in very good yields (**73-76**). We also applied our method for the direct synthesis of pharmaceutical drugs. The analog of Haloperidol (**77**) could be synthesized in 55% yield in one step (Fig. 4b). At last, we performed the

model reaction on 10 mmol scale under the sunlight. Compound **4** was isolated in high yield (2.8 g, 86%), which indicated a promising scale-up potential of this method using green energy (Fig. 4c).

**Mechanistic studies**. A number of experimental observations provide support for the proposed mechanism (Fig. 5). First, we synthesized the enamine (**78**) and subjected it to the reaction conditions. The desired product (**4**) was obtained in 88% yield, suggesting that the in-situ formed enamine was the reaction intermediate for the three-component reaction. Then we performed the radical addition of ethyl 2-mercaptoacetate (**3**) to enamine (**79**) in $D_2O$/DCM. We were able to isolate the deuterated product (**80**) in 92% yield with 90% deuteration, which clearly demonstrated that radical intermediate **II** was formed and then intercepted though deuterium atom transfer during the reaction (for a detailed account of the mechanism study, see Supplementary Fig. 5 and Fig. 6).

## Discussion

Compared with the classical Mannich reaction, which utilizes non-enolizable aldehyde and provides β-amino-carbonyl product via an ionic pathway, this three-component radical homo Mannich reaction makes use of enolizable aldehyde and affords γ-amino-carbonyl scaffold though a radical process. We anticipate that this method will be complementary to the classical Mannich reaction and it will simplify the design and construction of complex tertiary alkyl amine for chemists in both academia and industry.

## Methods

**General procedure for three-component radical homo Mannich reaction**. To a 10-mL oven-dried round bottomed flask were added 4 Å molecular sieves (1 g) and triphenylphosphine (393 mg, 1.5 mmol). The flask was degassed three times and protected with $N_2$ before anhydrous DCM (6 ml) was added. Amine (1 mmol), aldehyde/ketone (1.5 mmol), thiol (1.5 mmol), and DTBP (292 mg, 2 mmol, 0.37 ml) were added into the reaction mixture in order by micro-syringe. The reaction was stirred and irradiated using two 40 W household CFL bulbs (6 cm away, to keep the reaction at room temperature) at room temperature for 10 h. When the reaction was complete, EtOAc was added (20 ml). The mixture was dried with sodium sulfate, filtered and concentrated. The residue was purified by column chromatography on silica to give the product.

## Data availability

Materials and methods, experimental procedures, useful information, mechanistic studies, optimization studies, [1]H NMR spectra, [13]C NMR spectra, and mass spectrometry data are available in the Supplementary Information. Raw data are available from the corresponding author on reasonable request.

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

## Acknowledgements
We thank Shanghai Jiao Tong University for financial support.

## Author contributions
Z.S. conceived the project. S.S., W.Q, P.M., R.L. and X.L. performed all experiments. All the authors analyzed the results. Z.S. and S.S. wrote the manuscript.

## Competing interests
The authors declare no competing interests.
