## [Peer Review File · Nature Communications]

REVIEWER COMMENTS

Reviewer #1 (Remarks to the Author):

Suan and coworkers described the three-component radical homo Mannich reaction driven by visible-light. This manuscript represents an excellent research and describe a very useful methodology to prepare highly functionalized tertiary amines. In my opinion this methodology can be accepted for publication in Nature Communications after Major revisions, for several reasons:

-The scope of the reaction is very extensive and complete, with a wide range of substrates. Also the application to the functionalization of drugs is correct. However, the authors should also try to apply their methodology, with some transformations of the products, for the synthesis of intermediates or core structures of natural products or pharmaceutical drugs.

-In the references the number of reviews is very high (25/35 references), however the Mannich reviews are quite old (1998 and 2004). The authors should include more recent reviews about Mannich reactions. The authors should include as well examples of functionalization of enamines with radicals through photochemistry.

There is a paper that should be cited in the general Figure 1 and in the text because is related to the chemistry presented in the article. The example of Gaunt group, Chem. Sci. 2020, DOI:10.1039/D0SC04853D. This article is very similar, although in the present manuscript, due to the importance of the Mannich reaction and description of a homo Mannich reaction, deserves a publication in Nat. Chem.

- The SI is quite complete. However, there are several mistakes that should be fixed. For example:

-In compounds 6, 58 and 72 the authors should provide the C-F coupling constant in the ¹³C NMR.

For example in compound 6, there are more aromatic carbons because the authors have take a look of the doublet due to the coupling constant between C and F. The authors should provide the ¹⁹F NMR of these compounds.

-The authors should assign which signals are from the major diastereoisomer and the minor diastereoisomer, in the compounds where there is dr.

-The authors should check carefully the compounds with diastereotopic groups, because there are mistakes in the description of the ¹H-NMR. For example, in compound 52 the signal at 3.38 (d, J = 8.0 Hz, 6H) is not correct. The two MeO of the compound are diastereotopics. Every Meo is a singlet so should be 3.39 (s, 3H) and 3.37 (s, 3H). Please check all the compounds with a chiral center, where there is the possibility of diastereotopic groups.

Reviewer #2 (Remarks to the Author):

Zhankui Sun et. al. has developed three components radical homo Mannich reaction between desulfurization of thiol to generate electrophilic radical with enolizable aldehydes and amines under mild conditions. Authors utilized enolizable aldehydes in this homo Mannich reaction which expands the scope of this methodology to synthesize various γ -amino carbonyl compounds. This reaction proceeds efficiently with amine-containing alkenes, halides, cyano and esters without affecting those functional groups. Along with different enolizable aldehydes, and various cyclic and aromatic ketones also underwent with these reaction conditions. Aliphatic enolizable aldehydes, and various thiols were also tested to afford the corresponding products in good yields. In addition, authors also performed 10.0 mmol scale reaction and complex drug amines also tolerated with this reaction condition. Control experiments also prove this radical reaction pathway. With all this merit, I recommend this article for the publication in Nature Communications. The authors please address the issues listed below.

1. Have you performed the homo Mannich reaction with α,α' -both enolizable unsymmetric ketones as enamine components? It would be beneficial information for readers in terms of regioselectivity.
2. α -Sulfide ketones were also readily accommodated (68 - 70), change this into α -substituted sulfide ketones were also readily accommodated (69 - 70).
3. In figure 2 c and wherever it applicable, give a super script to dr values and mention as the dr value was based on ¹H NMR spectra
4. Add this reference also for desulfurization (reference no 34) Chem. Commun., 2019, 55, 10583--10586.

5. Besides, the thiol itself is an excellent hydrogen atom transfer reagent to intercept radical intermediate II. Please add this reference here, Glass, R. S., Sulfur Radicals and Their Application. Topics in Current Chemistry 2018, 376:22 (doi.org/10.1007/s41061-018-0197-0)
 6. In SI at page no 1, since all the page numbers were given as S1, S2, S3,..etc, here also change the number into S2, S3, S6.. and so on
 7. Add the content no 9 before the spectral data at page no S50
 8. Please mention double equivalent Carbon atoms in ¹³C NMR
 9. For compound 14 in page S14 C₂₄H₃₁NO₄ it is C₂₄H₃₂NO₄
 10. For compound 15 in SI page no S14, please correct the title of the compound in to ethyl 3-benzyl-4-(benzyl(3-ethoxy-3-oxopropyl)amino)butanoate and change the substrate name ethyl 3-(methylamino)propanoate this wont give the expected compound.
 11. Did you utilize K₂CO₃ to synthesis compound 37 (page no S25). If yes means leave it, no means correct it.
 12. Check the substrate name of the compound no 72 page no S42, the correct one is N-methyl-3-phenyl-3-(4-(trifluoromethyl)phenoxy)propan-1-amine
 13. For Compound d, ¹³C NMR spectra were missing (page no S131); only data was given at page no S4. Please include the ¹³C NMR spectrum.
- Some of the below spectrums given integral value and data entered are mismatched. Please go through this issue.
- For compound 38 in the SI page no 84, in the spectrum it was given 6.78 equivalent at 2.28-2.00 ppm. But in the data, it is accounted as 4H (page no S25).
 - For compound 40 in the SI page no 86, in the spectrum it was given 7.28 equivalent at 2.34-2.04 ppm. But in the data, it is accounted as 4H (page no S26).
 - For compound 41 in the SI page no 87, in the spectrum it was given 6.95 equivalent at 2.29-2.04 ppm. But in the data, it is accounted as 4H (page no S27).
 - For compound 46 in the SI page no 92, in the spectrum it was given 8.22 equivalent at 2.49-2.06 ppm. But in the data, it is accounted as 5H (page no S29).
 - For compound 67 in the SI page no 115, in the spectrum it was given 7.19 equivalent at 2.46-2.03 ppm. But in the data, it is accounted as 4H (page no S39).

Response to referees

Reviewer #1 (Remarks to the Author):

Suan and coworkers described the three-component radical homo Mannich reaction driven by visible-light. This manuscript represents an excellent research and describe a very useful methodology to prepare highly functionalized tertiary amines. In my opinion this methodology can be accepted for publication in Nature Communications after Major revisions, for several reasons:

-The scope of the reaction is very extensive and complete, with a wide range of substrates. Also the application to the functionalization of drugs is correct. However, the authors should also try to apply their methodology, with some transformations of the products, for the synthesis of intermediates or core structures of natural products or pharmaceutical drugs.

Thanks for the suggestions. We applied this methodology for the synthesis of an analogue of Haloperidol. This was included in the paper as figure 4b

-In the references the number of reviews is very high (25/35 references), however the Mannich reviews are quite old (1998 and 2004). The authors should include more recent reviews about Mannich reactions. The authors should include as well examples of functionalization of enamines with radicals through photochemistry.

There is a paper that should be cited in the general Figure 1 and in the text because is related to the chemistry presented in the article. The example of Gaunt group, Chem. Sci. 2020, DOI:10.1039/D0SC04853D. This article is very similar, although in the present manuscript, due to the importance of the Mannich reaction and description of a homo Mannich reaction, deserves a publication in Nat. Chem.

Thanks for the suggestions. The following papers have been cited.

9. Meyer, C. C., Ortiz, E. & Krische, M. J. Catalytic Reductive Aldol and Mannich Reactions of Enone, Acrylate, and Vinyl Heteroaromatic Pronucleophiles. Chem. Rev. 120, 3721-3748 (2020).
35. Kolahdouzan, K., Kumar, R. & Gaunt, M. J. Visible-light mediated carbonyl trifluoromethylative amination as a practical method for the synthesis of β -trifluoromethyl tertiary alkylamines Chem. Sci., 11, 12089-12094 (2020).

- The SI is quite complete. However, there are several mistakes that should be fixed. For example:
-In compounds 6, 58 and 72 the authors should provide the C-F coupling constant in the ¹³C NMR. For example in compound 6, there are more aromatic carbons because the authors have take a look of the doublet due to the coupling constant between C and F. The authors should provide the ¹⁹F NMR of these compounds.

Thanks for the suggestions. We have corrected them in the new SI.

-The authors should assign which signals are from the major diastereoisomer and the minor diastereoisomer, in the compounds where there is dr.

Thanks for the suggestions. We have tried our best to assign the signals. But, sometimes, it's really difficult. I hope the reviewer could understand. Thanks.

-The authors should check carefully the compounds with diastereotopic groups, because there are mistakes in the description of the ¹H-NMR. For example, in compound 52 the signal at 3.38 (d, J = 8.0 Hz, 6H) is not correct. The two MeO of the compound are diastereotopics. Every Meo is a singlet so should be 3.39 (s, 3H) and 3.37 (s, 3H). Please check all the compounds with a chiral center, where there is the possibility of diastereotopic groups.

Thanks for pointing out. We have corrected the mistakes.

Reviewer #2 (Remarks to the Author):

Zhankui Sun et. al. has developed three components radical homo Mannich reaction between desulfurization of thiol to generate electrophilic radical with enolizable aldehydes and amines under mild conditions. Authors utilized enolizable aldehydes in this homo Mannich reaction which expands the scope of this methodology to synthesize various γ -amino carbonyl compounds. This reaction proceeds efficiently with amine-containing alkenes, halides, cyano and esters without affecting those functional groups. Along with different enolizable aldehydes, and various cyclic and aromatic ketones also underwent with these reaction conditions. Aliphatic enolizable aldehydes, and various thiols were also tested to afford the corresponding products in good yields. In addition, authors also performed 10.0 mmol scale reaction and complex drug amines also tolerated with this reaction condition. Control experiments also prove this radical reaction pathway. With all this merit, I recommend this article for the publication in Nature Communications. The authors please address the issues listed below.

1. Have you performed the homo Mannich reaction with α,α' -both enolizable unsymmetric ketones as enamine components? It would be beneficial information for readers in terms of regioselectivity.

Thanks for the suggestions. We have compound 53, which is derived from unsymmetric ketones. As you could see, the regioselectivity is favor of the less hindered side.

We also synthesized the following compound. Although the yield is only 29% due to the steric

hindrance, but the regioselectivity is quite clear. This is included in the new manuscript as compound 52.

2. α -Sulfide ketones were also readily accommodated (68 - 70), change this into α -substituted sulfide ketones were also readily accommodated (69 - 70).

Thanks for pointing out. We have corrected the mistakes.

3. In figure 2 c and wherever it applicable, give a super script to dr values and mention as the dr value was based on ¹H NMR spectra

Thanks for pointing out. We have added the super script.

4. Add this reference also for desulfurization (reference no 34) *Chem. Commun.*, 2019, 55, 10583--10586.

Thanks for pointing out. We have added the reference.

36 Qin, Q., Wang, W., Zhang, C., Song, S. & Jiao, N. A metal-free desulfurizing radical reductive C-C coupling of thiols and alkenes. *Chem. Commun.*, **55**, 10583-10586 (2019).

5. Besides, the thiol itself is an excellent hydrogen atom transfer reagent to intercept radical intermediate II. Please add this reference here, Glass, R. S., *Sulfur Radicals and Their Application. Topics in Current Chemistry* 2018, 376:22 (doi.org/10.1007/s41061-018-0197-0)

Thanks for pointing out. We have added the reference.

38 Glass, R. S. *Sulfur Radicals and Their Application. Top Curr Chem*, **376**, 22 (2018).

6. In SI at page no 1, since all the page numbers were given as S1, S2, S3,..etc, here also change the number into S2, S3, S6.. and so on

Thanks for pointing out. We made the changes.

7. Add the content no 9 before the spectral data at page no S50

Thanks for pointing out. We made the changes.

8. Please mention double equivalent Carbon atoms in ¹³C NMR

Thanks for pointing out. We made the changes.

9. For compound 14 in page S14 C₂₄H₃₁NO₄ it is C₂₄H₃₂NO₄

Thanks for pointing out. We made the changes.

10. For compound 15 in SI page no S14, please correct the title of the compound in to ethyl 3-benzyl-4-(benzyl(3-ethoxy-3-oxopropyl)amino)butanoate and change the substrate name ethyl 3-(methylamino)propanoate this wont give the expected compound.

Thanks for pointing out. We made the corrections.

11. Did you utilize K₂CO₃ to synthesis compound 37 (page no S25). If yes means leave it, no means correct it.

Yes, we used K₂CO₃ to synthesis compound 37. Acetaldehyde is very reactive. We tried a lot and found out adding K₂CO₃ helped to deliver the product.

12. Check the substrate name of the compound no 72 page no S42, the correct one is N-methyl-3-phenyl-3-(4-(trifluoromethyl)phenoxy)propan-1-amine

Thanks for pointing out. We made the corrections.

13. For Compound d, ¹³C NMR spectra were missing (page no S131); only data was given at page no S4. Please include the ¹³C NMR spectrum.

Thanks for pointing out. We added the ¹³C NMR spectrum

Some of the below spectrums given integral value and data entered are mismatched. Please go through this issue.

- For compound 38 in the SI page no 84, in the spectrum it was given 6.78 equivalent at 2.28-2.00 ppm. But in the data, it is accounted as 4H (page no S25).
- For compound 40 in the SI page no 86, in the spectrum it was given 7.28 equivalent at 2.34-2.04 ppm. But in the data, it is accounted as 4H (page no S26).
- For compound 41 in the SI page no 87, in the spectrum it was given 6.95 equivalent at 2.29-2.04 ppm. But in the data, it is accounted as 4H (page no S27).
- For compound 46 in the SI page no 92, in the spectrum it was given 8.22 equivalent at 2.49-2.06 ppm. But in the data, it is accounted as 5H (page no S29).
- For compound 67 in the SI page no 115, in the spectrum it was given 7.19 equivalent at 2.46-2.03 ppm. But in the data, it is accounted as 4H (page no S39).

Thanks for pointing out. We made the corrections.